# Natural Course of COVID-19 and Independent Predictors of Mortality

**DOI:** 10.3390/biomedicines11030939

**Published:** 2023-03-17

**Authors:** Luana Orlando, Gianluca Bagnato, Carmelo Ioppolo, Maria Stella Franzè, Maria Perticone, Antonio Giovanni Versace, Angela Sciacqua, Vincenzo Russo, Arrigo Francesco Giuseppe Cicero, Alberta De Gaetano, Giuseppe Dattilo, Federica Fogacci, Maria Concetta Tringali, Pierpaolo Di Micco, Giovanni Squadrito, Egidio Imbalzano

**Affiliations:** 1Department of Clinical and Experimental Medicine, University of Messina, 98125 Messina, Italy; 2Department of Medical and Surgical Sciences, University Magna Græcia of Catanzaro, 88100 Catanzaro, Italy; 3Department of Medical Translational Sciences, Division of Cardiology, Monaldi Hospital, University of Campania “Luigi Vanvitelli”, 80100 Naples, Italy; 4IRCCS Policlinico S. Orsola—Malpighi, Hypertension and Cardiovascular risk Research Center, DIMEC, University of Bologna, 40100 Bologna, Italy; 5Department of Medicine, PO Santa Maria delle Grazie Pozzuoli, 80100 Naples, Italy

**Keywords:** NT-pro-BNP, prothrombin time, PaO_2_/FiO_2_, COVID-19, biomarkers, SARS-CoV-2, coronavirus disease

## Abstract

Background: During the SARS-CoV-2 pandemic, several biomarkers were shown to be helpful in determining the prognosis of COVID-19 patients. The aim of our study was to evaluate the prognostic value of N-terminal pro-Brain Natriuretic Peptide (NT-pro-BNP) in a cohort of patients with COVID-19. Methods: One-hundred and seven patients admitted to the Covid Hospital of Messina University between June 2022 and January 2023 were enrolled in our study. The demographic, clinical, biochemical, instrumental, and therapeutic parameters were recorded. The primary outcome was in-hospital mortality. A comparison between patients who recovered and were discharged and those who died during the hospitalization was performed. The independent parameters associated with in-hospital death were assessed by multivariable analysis and a stepwise regression logistic model. Results: A total of 27 events with an in-hospital mortality rate of 25.2% occurred during our study. Those who died during hospitalization were older, with lower GCS and PaO_2_/FiO_2_ ratio, elevated D-dimer values, INR, creatinine values and shorter PT (prothrombin time). They had an increased frequency of diagnosis of heart failure (*p* < 0.0001) and higher NT-pro-BNP values. A multivariate logistic regression analysis showed that higher NT-pro-BNP values and lower PT and PaO_2_/FiO_2_ at admission were independent predictors of mortality during hospitalization. Conclusions: This study shows that NT-pro-BNP levels, PT, and PaO_2_/FiO_2_ ratio are independently associated with in-hospital mortality in subjects with COVID-19 pneumonia. Further longitudinal studies are warranted to confirm the results of this study.

## 1. Introduction

Two years after the outbreak of the coronavirus disease 2019 (COVID-19) pandemic, scientific progress and new vaccines has slowed mortality rates worldwide. However, in the first months, when the pandemic was announced in March 2020, the incidence of hospitalizations and mortality was very high. Patients with cardiovascular comorbidities, type 2 diabetes mellitus (T2DM), chronic kidney disease, cancer, pulmonary diseases, obesity, and those that are elderly were determined to have the most severe prognosis with the most significant mortality [1].

Although the main manifestations of COVID-19 are respiratory, septic and thromboembolic, other conditions such as brain damage, male infertility and pregnancy complications have been associated with severe acute respiratory syndrome coronavirus 2 (SARS-CoV-2), [2,3].

Among pregnancy complications, hypertensive disorders including preeclampsia and preeclampsia-like syndrome are associated with a dysregulated activity of the renin–angiotensin–aldosterone system. This disorder is common with COVID-19 and could be explained by high aldosterone levels causing hypertension and inflammation [4]. 

The cardiovascular system appears to be directly affected by numerous interactions with SARS-CoV-2, as suggested by the evidence of myocardial damage in children [5,6] and adult patients [7]. Several studies have investigated the possible correlation between cardiac biomarkers and the severity of COVID-19 disease [8,9,10] in order to identify one or more cut-off values to stratify patients for disease severity. Most of them show that B-type natriuretic peptide (BNP) as a predictive marker of the severity of COVID-19 disease [11]. Thus, genes and signaling pathways involved in the myocardium infected by SARS-CoV-2 were subsequently identified highlighting the mechanisms underlying cardiac damage [12]. Furthermore, numerous studies have revealed a key role for the angiotensin converting enzyme-2 (ACE2) receptor, whose physiological balance is altered by SARS-CoV-2, by activating angiotensin II (Ang II)/Ang II receptor type 1 (AT1R) pathways and by leading to severe disease complications [13,14]. The aim of our study was to evaluate the association between N-terminal fragment B-type natriuretic peptide (NT-pro-BNP) values and disease severity during the late phase of the pandemic, discovering a possible cut-off value to stratify the disease progression in COVID-19 patients.

## 2. Patients and Methods

This was a single center, retrospective, observational study. A total of 107 patients with a confirmed diagnosis of COVID-19, consecutively admitted between June 2022 and January 2023, at the COVID Division of the Hospital of the University of Messina, were enrolled in the study. The hospital is a large structure classified as a specialized second level hospital center serving an area of approximately 600,000 residents [15]. The diagnosis of SARS-CoV-2 infection was defined by using a Real-Time Polymerase Chain Reaction (RT-PCR) test, performed with oligonucleotide primers and probes drawn on conserved regions of the SARS-CoV-2 genome on biological material, collected by nasopharyngeal and oropharyngeal swabs. The swabs were repeated every 2–4 days to assess viral load clearance and the negativity of the test. 

An anamnestic investigation, a complete objective examination, chest X-rays, and laboratory tests were recorded from each patient at hospital admission. The length of hospitalization was calculated from the date of admission to the hospital (in most cases corresponding to the date of diagnosis of COVID-19) to either discharge or in-hospital death. The presence of pre-existing cardiovascular diseases (e.g., arterial hypertension, coronary heart disease, heart failure) were recorded. The New York Heart Association (NYHA) classification was used to stratify patients with heart failure [16]. Treatments with Angiotensin-converting enzyme (ACE) inhibitors and Renin-Angiotensin-Aldosterone System (RAAS) inhibitors were recorded. Symptoms and signs of deep vein thrombosis (DVT) were sought in each patient both clinically and by performing bilateral compression ultrasonography (CUS), with the possible localization of venous thrombi and/or the development of a pulmonary embolism. Some patients underwent chest computed tomography (CT), including CT angiography or CT of the brain, abdomen, or limbs if required. The use of drugs for thromboprophylaxis was recorded, and it was specified whether the subject was treated with enoxaparin sodium or fondaparinux, vitamin K antagonists (VKAs), or non-vitamin K antagonists (NOACs). The partial pressure of oxygen (PaO_2_) values, the fractional concentration of oxygen in inspired air (FiO_2_) values, and the PaO_2_/FiO_2_ ratio were recorded at admission, as well as the oxygen-therapy required and the need for mechanical ventilation. The mean arterial pressure (MAP) was calculated at admission with the following formula: diastolic pressure +1/3 (systolic pressure—diastolic pressure). 

The following laboratory tests were performed for each patient at admission: serum concentration of D-dimer, prothrombin time (PT), international normalized ratio (INR), fibrinogen, platelet count, serum creatinine, serum total bilirubin, NT-pro-BNP. The normal range of NT-pro-BNP in our laboratory was 0–300 ng/L. 

Admission to the Intensive Care Unit (ICU) occurred in some patients due to the deterioration in vital functions and/or a worsening of the laboratory profile and/or instrumental evidence of signs of acute complications. The primary outcome of this study was in-hospital mortality, which was defined as death occurring during hospitalization. 

No subject included in the study was asked for informed consent, since it was a non-interventional study and proposed as an anonymous retrospective analysis.

The study was conducted in accordance with the Declaration of Helsinki, and was approved by the or Ethics Committee of the University of Messina (protocol code 41-20;04/05/2020).

### Statistical Analysis

All statistical analyses were performed by IBM SPSS software (Version 26.0, SPSS, USA). The numerical data were expressed as mean and standard deviation (SD), whereas the categorical variables as a number and percentage. Examined continuous variables did not present a normal distribution as verified by a Kolmogorov–Smirnov test, and the non-parametric approach was consequently used. Accordingly, to compare the variables divided for survivors and deceased patients during hospitalization, the Mann–Whitney test was used for numerical variables. For categorical variables, the comparison between two groups was performed by Pearson’s chi-square test. Univariate logistic regression analysis was used to identify predictors of death. Variables with a *p* value < 0.05 at univariate analysis were included in the multivariate logistic model. The results were expressed as odds ratio (OR) with a relative 95% confidence interval (CI) and *p* value. Furthermore, a stepwise regression model was designed to evaluate the potential independent predictors of mortality. The statistical significance level was set at a *p* value < 0.05.

## 3. Results

### 3.1. Patients’ Characteristics

One hundred and seven COVID-19 patients [50 (46.7%) males/57 (53.3%) females with a mean age of 71.5 ± 16.1 years] that were consecutively hospitalized were enrolled in the study. The mean length of stay was 28.5 ± 17.3 days. During the hospitalization, four patients (3.7%) were transferred to the ICU and 27 (25.2%) died. Demographic, clinical, and biochemical characteristics of the study population are reported in Table 1. Thirty-four patients (31.8%) required oxygen-therapy at admission, and eight (7.5%) were treated with mechanical ventilation. The mean PaO_2_/FiO_2_ was 74.2 ± 20.2 mmHg, whereas the mean percentage of FiO_2_ required was 26.4 ± 11.6. Sixty patients (56.1%) had a diagnosis of arterial hypertension, 16 (15%) had ischemic heart disease, and 55 (51.4%) had heart failure. Specifically, 7/55 patients (12.7%) were in NYHA class I, 14 (25.5%) were in NYHA class II, 14 (25.5%) were in NYHA class III, and 20 (36.4%) were in in NYHA class IV. Nineteen patients (17.8%) were under treatment with ACE inhibitors, and 31 (29%) were treated with RAAS inhibitors. At admission, the mean NT-pro-BNP value was 1954 ± 4941 pg/mL. During the hospitalization, serial compression ultrasonography (CUS) was performed in 50 patients (46.7%), and 13 patients (12.1%) underwent a chest CT scan. Only in three cases (2.8%) the instrumental investigation confirmed a diagnosis of deep venous thrombosis (DVT). The mean serum D-dimer levels recorded at admission was 1455 ± 1252 ng/mL. In particular, 27 patients (25.2%) had normal values (0–500 ng/mL), 29 (27.1%) 29 (27.1%) had mildly increased concentrations (500–1000 ng/mL), 35 (32.7%) had moderately elevated levels (1000–4000 ng/mL), and 16 (15%) were severely elevated (>4000 ng/mL). During the hospitalization, 103/107 (96.3%) patients underwent pharmacological thromboprophylaxis: 94/103 (87.9%) were treated with enoxaparin or fondaparinux, 3/103 (2.8%) were treated with VKAs, and 6/103 (5.6%) were treated with NOACs (rivaroxaban, apixaban, or edoxaban). 

### 3.2. Comparison between Survivors and Deceased COVID-19 Patients

Next, we analyzed the study population after dividing them according to the outcome of interest as deceased (n = 27) and survivors (n = 80). As reported in Table 1, no statistically significant differences were observed concerning sex, BMI, MAP, presence of arterial hypertension, ischemic heart disease, treatment with ACE-inhibitors, or RAAS-inhibitors, bilirubin values, platelet count, fibrinogen levels, the presence of DVT signs (assessed by CUS or CT), and treatment with anticoagulants. 

On the contrary, deceased patients were older, with lower GCS and a shorter length of hospitalization and more severe pulmonary involvement, as evidenced by the difference in PaO_2_/FiO_2_ values. Deceased patients had higher D-dimer values, shorter prothrombin time, higher INR and higher creatinine values at admission compared to survivors. Moreover, patients who died significantly differed from survivors with regard to the presence of heart failure with higher NYHA classes and higher NT-pro-BNP values (Table 1 and Table 2). 

### 3.3. Analysis of Independent Predictors of In-Hospital Death

We investigated the possible association between all parameters collected at admission and in-hospital mortality. Since age was significantly different between the groups, the initial univariate logistic regression was corrected for age. Upon univariate logistic regression analysis, mortality was significantly associated with lower PaO_2_/FiO_2,_ lower PT values (*p* = 0.004), higher NT-pro-BNP values (*p* = 0.001), and lower Glasgow Coma Scale classes (*p* = 0.001). A multivariate stepwise logistic regression analysis showed that higher NT-pro-BNP values, lower PT values, and lower PaO_2_/FiO_2_ at admission were independent predictors of mortality during hospitalization. (Table 3).

## 4. Discussion

COVID-19 is known as a respiratory illness; however, current research suggests that SARS-CoV 2 has a direct cardiomyocyte tropism. As a result, myocarditis, the development of left ventricular systolic and diastolic dysfunction, and myocardial fibrosis might occur [17]. Due to the predominance and severity of pulmonary involvement, these individuals may also have right ventricular dysfunction. The inflammatory state caused by the action of the virus [18] induces the infiltration of the myocardium by inflammatory cells (monocytes, macrophages, lymphocytes, neutrophils), leading to myocarditis and consequently heart failure [19,20]. Myocarditis, as a result of an autoimmune response induced by the virus, could progress in structural cardiomyopathy [21], and could be associated with a poor prognosis when complicated by heart failure and left ventricular dysfunction [22,23]. This is particularly relevant considering that a recent meta-analysis compared fulminant myocarditis associated with COVID-19 infection versus COVID-19 vaccination. While this study reported a similarly high mortality rate and no differences in most biopsies/autopsies between these conditions, COVID-19 fulminant myocarditis was associated with a more aggressive course with more severe hemodynamic decompensation and more cardiac arrests [24]. 

This is particularly relevant, since an excess of myocardial injury may cause an increase in several cardiac biomarkers, such as NT-pro-BNP. NT-pro-BNP is a quantitative plasma biomarker reflecting hemodynamic cardiac stress that is usually caused by volume or pressure overload, and therefore plays a central role in the diagnosis and management of heart failure and cardiac function. Yet, seeing as its concentrations increase immediately after myocardial damage and improves as a consequence of clinical improvement, it represents a predictor of an adverse outcome in acute myocardial damage [25]. Myocardial injury, inflammation, interaction with angiotensin converting enzyme 2 (ACE2), or impairment of cardiac function with acute heart failure may be responsible for higher circulating natriuretic peptides in COVID-19 patients. Yang et al. retrospectively analyzed 203 patients with a confirmed diagnosis of SARS-CoV-2 infection and definite outcomes (discharge or death), consisting of 145 patients who recovered and 58 patients who died. In their analysis, 53% of the deceased had elevated NT-pro-BNP levels [26]. According to a recent meta-analysis involving 2248 patients, with the majority belonging to the early COVID-19 epidemic in China, NT-pro-BNP evaluation may help differentiate high-risk individuals [27]. Furthermore, as noted by Gao et al., patients with severe COVID-19 who have high NT-pro-BNP levels are more likely to be older, have greater levels of systemic inflammatory markers and heart damage markers, and have a poorer cumulative survival rate. After adjusting for relevant confounders, NT-pro-BNP was demonstrated to be an independent risk factor for in-hospital death in patients with severe COVID-19 [28]. According to other studies, it has been demonstrated that NT-pro-BNP levels were eight times higher at the time of hospitalization in deceased patients versus survivors. Therefore, despite the fact that cardiac injury is a common condition among hospitalized patients with COVID-19, high NT-pro-BNP levels were associated with a higher risk of in-hospital mortality [29]. Furthermore, numerous reports support the predictive role of NT-pro-BNP in the severity of the disease [30,31] in patients with and without pre-existing heart failure [32]. In line with previous studies, our results confirm that high NT-pro-BNP levels at admission increase the risk of in-hospital mortality and are very strong and independent indicators of mortality in COVID-19 patients. The routine measurements of cardiac biomarkers and especially NT-pro-BNP should be considered in COVID-19 patients. If confirmed by larger population studies, it might be possible to identify a cut-off value of this biomarker in order to stratify the population at higher risk of developing severe disease. The PaO_2_/FiO_2_ ratio is used to classify the severity of acute respiratory distress syndrome (ARDS), and is the most commonly used oxygenation index included in the sepsis management guidelines [33] and acute respiratory distress syndrome [34]. In a non-COVID-19 setting, PaO_2_/FiO_2_ is considered to be a predictor of unfavorable outcomes, as initially reported by Villar et al. [35], and subsequently verified in many additional studies, primarily from ICU settings [36,37]. Unexpectedly, despite being widely used in clinical practice, not many reports have previously investigated its ability to predict mortality in COVID-19 patients in critical care settings. In the context of COVID-19, the PaO_2_/FiO_2_ ratio has primarily been studied as an indicator of disease severity and as a predictor of mortality [38,39,40]. In this study, PaO_2_/FiO_2_ was an independent predictor of COVID-19 mortality. Therefore, our findings have the potential clinical relevance to support the use of a single PaO_2_/FiO_2_ ratio measurement at admission as an independent predictor of mortality. Prothrombin time (PT) levels are elevated in COVID-19 patients. Higher PT was noted in 183 individuals whose data were examined [41]. In addition, PT is reported to be higher in patients admitted to the ICU and in severe COVID-19 cases compared to those with mild disease [42,43].

## 5. Conclusions

In conclusion, our results suggest that there are multiple predictors of mortality among COVID-19 patients, and in particular our findings support that COVID-19 induces a systemic derangement involving not only the pulmonary system but also the coagulation cascade and the cardiovascular system. Thus, NT-pro-BNP, PaO_2_/FiO_2,_ and PT could potentially serve as independent risk factors for predicting death in COVID-19 patients. It is worthy of note that clinicians might want to consider the aforementioned indicators and take action to reduce the mortality of COVID-19. Eventually, future confirmatory studies in larger prospective cohorts might prove useful for patients stratification into risk classes. This would allow to allocate patients to the appropriate intensity of care according to disease severity and adapt treatment regimen to the multifaceted systemic aspects of the disease. 

## Figures and Tables

**Table 1 biomedicines-11-00939-t001:** Demographic and clinical characteristics of 107 COVID-19 patients included in the study.

Baseline Characteristics	Total (n = 107)	Survivors (n = 80)	Deceased (n = 27)	*p*
Male, n (%)	50 (46.7)	40 (50)	10 (37)	0.243
BMI, kg/m^2^	23.48± 3.06	23.57 ± 2.99	23.11 ± 3.36	0.356
ICU, n (%)	4 (3.7)	3 (3.8)	1 (3.7)	0.991
Age, years	71.51 ± 16.14	68.13 ± 16.21	81.56 ± 11.1	**<0.0001**
Length of stays	28.45 ± 17.31	32.56 ± 15.83	16.29 ± 15.95	0.476
CUS, n (%)	50 (46.7)	37 (46.3)	13 (48.1)	0.864
DVT confirmed, n (%)	3 (2.8)	1 (1.3)	2 (7.4)	0.094
CT scan, n (%)	13 (12.1)	10 (12.5)	3 (11.1)	0.849
Drug thromboprophylaxis n (%)	103 (96.3)	77 (96.3)	26 (96.3)	0.991
Anticoagulant drugs, n (%)				
No therapy	4 (3.7)	3 (3.8)	1 (3.7)	
LMWH	94 (87.9)	70 (87.5)	24 (88.9)	
AVKs	3 (2.8)	2 (2.5)	1 (3.7)	0.952
NOACs	6 (5.6)	5 (6.3)	1 (3.7)	
COVID-19 vaccination	19 (17.7)	13 (16.2)	6 (22.2)	0.482
Mechanical ventilation, n (%)	8 (7.5)	4 (5)	4 (14.8)	0.094
PaO_2_/FiO_2_ ratio	314 ± 115	342 ± 102	228 ± 113	<0.001
MAP, mmHg	87 ± 12.42	86.91 ± 11.98	87.25 ± 49.33	0.773
CCI	2.6 ± 1.3	2.3 ± 1.1	2.9 ± 1.7	0.334
Arterial hypertension, n (%)	60 (56.1)	42 (52.5)	18 (66.7)	0.200
Coronary heart disease, n (%)	16 (15.0)	11 (13.8)	5 (18.5)	0.548
ACE-Inhibitors, n (%)	19 (17.8)	14 (17.5)	5 (18.5)	0.905
RAAS, n (%)	31 (29.0)	22 (27.5)	9 (33.3)	0.563
Heart failure, n (%)	55 (51.4)	33 (41.3)	22 (81.5)	**<0.0001**
NYHA classes, n (%)				
Class I	7 (12.7)	6 (18.2)	1 (4.5)	
Class II	14 (25.5)	12 (36.4)	2 (9.1)	
Class III	14 (25.5)	9 (27.3)	5 (22.7)	**0.004**
Class IV	20 (36.4)	6 (7.5)	14 (63.6)	
GCS	13.54 ±2.45	14.08 ± 2.11	11.96 ± 2.73	**<0.0001**

All numerical parameters are expressed as mean and standard deviations (SD). Bold characters identify statistically significant results. ICU: Intensive Care Unit. CUS: Compression Ultrasonography. VKA: Vitamin K Antagonist. DVT: Deep Venous thrombosis. CT: Computer tomography. DOACs: Direct Oral Anticoagulants SD: Standard Deviation PT: Prothrombin Time. INR: International Normalized Ratio. GCS: Glasgow Coma Scale. MAP: Mean Arterial Pression. CCI: Charlson Comorbidity index; *p*: *p*-value.

**Table 2 biomedicines-11-00939-t002:** Biochemical characteristics of 107 COVID-19 patients included in the study.

	Baseline Characteristics	Survivors (n = 80)	Deceased (n = 27)	*p*
Creatinine, mg/dL	1.16 ± 1.22	0.93 ± 0.46	1.84 ± 2.91	**0.029**
Bilirubine, mg/dL	0.51 ± 0.27	0.49 ± 0.23	0.59 ± 0.35	0.182
NT-proBNP, pg/mL	1954 ± 4941	718 ± 1286	5616 ± 8712	**<0.0001**
D-dimer, ng/mL	1455 ± 1252	1276 ±1166	1988 ± 1367	**0.005**
D-Dimer classes, n (%)				
I (0–500 ng/mL)	27 (25.2%)	24 (30%)	3 (11.1%)	
II (500–1000 ng/mL)	29 (27.1%)	22 (27.5%)	7 (25.9%)	
III (1000–4000 ng/mL)	35 (32.7%)	25 (31.3%)	10 (37%)	
IV (>4000 ng/mL)	17 (15.0%)	9 (11.3%)	7 (25.9%)	
PT, seconds	81.1 ± 22.43	85.06 ± 19.61	69.29 ± 26.24	**0.002**
INR	1.31 ± 1.21	1.13 ± 0.30	1.84 ± 2.30	**0.001**
Platelet, n/mm^3^	207.738 ± 80785	202.375 ± 77.733	190,000 ± 88.863	0.335

All numerical parameters are expressed as mean and standard deviations (SD). Bold characters identify statistically significant results. PT, prothrombin time; INR: International Nationalized Ratio.

**Table 3 biomedicines-11-00939-t003:** Age-adjusted univariate and multivariate stepwise backward logistic regression analysis of predictive factors for in-hospital mortality in patients with COVID-19.

	Univariate Analysis			Multivariate Analysis		
	Odds Ratio	CI 95%	*p*	Odds Ratio	CI 95%	*p*
D-Dimer			0.087	-	-	-
PT	0.964	0.941–0.988	**0.004**	0.958	0.927–0.990	**0.010**
INR			0.130			
Creatinine			0.083			
PaO_2_/FiO_2_ ratio	0.991	0.985–0.996	**0.001**	0.987	0.981–0.994	**0.001**
GCS	0.677	0.544–0.842	**0.001**			0.128
Heart failure			0.147	-	-	-
NYHA classes			0.055			
NT-pro-BNP	1.000	1.000–1.001	**0.026**	1.001	1.000–1.001	**0.004**

PT: prothrombin time. GCS: Glasgow Coma Scale.

## Data Availability

The data presented in this study are available on request from the corresponding author.

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
