# Peer review of "Natural Course of COVID-19 and Independent Predictors of Mortality"

_biomedicines, 2023, doi:10.3390/biomedicines11030939_

Round 1

Reviewer 1 Report

This is an interesting and generally well written study. Only some points could be improved. In particular:

Introduction: a more detailed introduction on SARS-CoV-2 infection is due. In fact, it deserves to be mentioned that SARS-CoV-2 can also lead to non restiratory diseases such as Preeclampsia, male infertility and brain damage (as recently reviewed PMID: 35114008, 35943095, 32934351). This is an important point to add since it highlights the fact that the cytokine storm found in COVID-19 patients can also damage other organs. This is particularly true for preeclampsia where a specific marker to distinguish real preeclampsia from covid-19 symptoms (e.g. hypertension during pregnancy) is needed. (see PMID: 35943095). This is an important point since it can further highlight the interesting results found by the authors regarding NT-pro-BNP.

Did the authors evaluate if NT-pro-BNP levels were associated to patient age? In table 1 is reported that deceased patients were significantly older and in table 2 authors reported a significant increase of NT-pro-BNP levels in deceased patients. I know that age variant is eliminated in age-adjusted univariate and multivariate logistic regression analysis but it would be interesting reporting a possible link between NT-pro-BNP levels and patients age.

Author Response

ANSWERS TO REVIEWER 1

This is an interesting and generally well written study. Only some points could be improved. In particular:

We thank the reviewer for the important suggestions.

Introduction: a more detailed introduction on SARS-CoV-2 infection is due. In fact, it deserves to be mentioned that SARS-CoV-2 can also lead to non restiratory diseases such as Preeclampsia, male infertility and brain damage (as recently reviewed PMID: 35114008, 35943095, 32934351). This is an important point to add since it highlights the fact that the cytokine storm found in COVID-19 patients can also damage other organs. This is particularly true for preeclampsia where a specific marker to distinguish real preeclampsia from covid-19 symptoms (e.g. hypertension during pregnancy) is needed. (see PMID: 35943095). This is an important point since it can further highlight the interesting results found by the authors regarding NT-pro-BNP.

Indeed, the overlap between inflammation and hypertensive disorders has been added in the introduction as this is a very intriguing and deserves a specific mention in the manuscript.

Did the authors evaluate if NT-pro-BNP levels were associated to patient age? In table 1 is reported that deceased patients were significantly older and in table 2 authors reported a significant increase of NT-pro-BNP levels in deceased patients. I know that age variant is eliminated in age-adjusted univariate and multivariate logistic regression analysis but it would be interesting reporting a possible link between NT-pro-BNP levels and patients age.

This is a very important suggestion regarding age and NT-pro-BNP levels and it remains a very complex area to explore. Unfortunately, we observed no significant correlation between these two variables.

Reviewer 2 Report

A very interesting study, a very useful approach for daily clinical practice.

I believe it is necessary to mention how many patients had active neoplasia at the time of admission, it is a factor directly related to the unfavorable evolution.

The characteristics of the hospital, size, population served, if it is a monograph of any pathology, etc. should be explained. and the Charlson index to globally understand the comorbidities of the patients analyzed.

It is necessary to mention the vaccination percentage, at least in the general population of the area, given the high mortality described.

An interesting point to review would be the relationship between vaccines and cardiac involvement in adults, since there are more cases described in young people, probably for a new article.

Author Response

ANSWERS TO REVIEWER 2

A very interesting study, a very useful approach for daily clinical practice.

We thank the reviewer for the interesting comments

I believe it is necessary to mention how many patients had active neoplasia at the time of admission, it is a factor directly related to the unfavorable evolution.

We carefully reviewed the study population and we found no patients with active neoplasia among them.

The characteristics of the hospital, size, population served, if it is a monograph of any pathology, etc. should be explained. and the Charlson index to globally understand the comorbidities of the patients analyzed.

The characteristics of the hospital, size and population served have been added in the materials and methods section as well as the Charlson comorbidity index

It is necessary to mention the vaccination percentage, at least in the general population of the area, given the high mortality described.

We carefully reviewed the study population and we reported the COVID-19 vaccination rate.

An interesting point to review would be the relationship between vaccines and cardiac involvement in adults, since there are more cases described in young people, probably for a new article.

This is indeed an interesting point to mention, we modified the discussion accordingly.